# Safety Evaluation of Tadalafil Treatment for Fetuses with Early-Onset Growth Restriction (TADAFER): Results from the Phase II Trial

**DOI:** 10.3390/jcm8060856

**Published:** 2019-06-15

**Authors:** Shintaro Maki, Hiroaki Tanaka, Makoto Tsuji, Fumi Furuhashi, Shoichi Magawa, Michiko K. Kaneda, Masafumi Nii, Kayo Tanaka, Eiji Kondo, Satoshi Tamaru, Toru Ogura, Yuki Nishimura, Masayuki Endoh, Tadashi Kimura, Tomomi Kotani, Akihiko Sekizawa, Tomoaki Ikeda

**Affiliations:** 1Department of Obstetrics and Gynecology, Mie University Graduate School of Medicine, 2-174 Edobashi, Tsu, Mie 514-8507, Japan; h_tanaka@med.miyazaki-u.ac.jp (H.T.); tototo.9696tj@gmail.com (M.T.); fumi19860812@gmail.com (F.F.); shoichimagawa@yahoo.co.jp (S.M.); michi_k_1019@yahoo.co.jp (M.K.K.); doaldosleeping@yahoo.co.jp (M.N.); tagami.t.ky@gmail.com (K.T.); eijikon@clin.medic.mie-u.ac.jp (E.K.); t-ikeda@clin.medic.mie-u.ac.jp (T.I.); 2Clinical Research Support Center, Mie University Hospital, 2-174 Edobashi, Tsu, Mie 514-8507, Japan; sato.tamaru@gmail.com (S.T.); t-ogura@clin.medic.mie-u.ac.jp (T.O.); ynishimura@clin.medic.mie-u.ac.jp (Y.N.); 3Department of Obstetrics and Gynecology, Osaka University Graduate School of Medicine, 2-15 Yamadaoka, Suita, Osaka 565-0871, Japan; endo@gyne.med.osaka-u.ac.jp (M.E.); tadashi@gyne.med.osaka-u.ac.jp (T.K.); 4Department of Obstetrics and Gynecology, Nagoya University Graduate School of Medicine, 65 Tsurumai-cho, Showa-ku, Nagoya 466-8560, Japan; itoto@med.nagoya-u.ac.jp; 5Department of Obstetrics and Gynecology, Showa University Graduate School of Medicine, 1-5-8, Hatanodai, Shinagawa-ku, Tokyo 142-8666, Japan; sekizawa@med.showa-u.ac.jp

**Keywords:** fetal growth restriction, tadalafil, phosphodiesterase 5 inhibitor

## Abstract

Tadalafil is a phosphodiesterase 5 (PDE5) inhibitor with a long half-life, high selectivity, and rapid onset of action. Because the safety of using PDE5 inhibitors as therapeutic agents for fetal growth restriction (FGR) has been a problem worldwide, this paper primarily focuses on the safety assessments performed in the Tadalafil Treatment for Fetuses with Early-Onset Growth Restriction (TADAFER) II population. Neonatal and maternal adverse events were analyzed, in addition to fetal, neonatal, and infant death cases, six months after stopping the trial. Eighty-nine pregnant women with FGR were studied between September 2016 and March 2018 (45 and 44 in the tadalafil and conventional treatment groups, respectively). Seven (16%) deaths (four fetal, one neonatal, and two infant) in the control group, whereas only one neonatal death occurred in the tadalafil group. Although headache, facial flushing, and nasal hemorrhage occurred more frequently in the tadalafil group, these symptoms were Grade 1 and transient. In conclusion, this trial showed that tadalafil decreased the fetal and infant deaths associated with FGR. This is thought to be primarily due to pregnancy prolongation. Further studies are warranted to evaluate the efficacy of tadalafil in treating early-onset FGR.

## 1. Introduction

Fetal growth restriction (FGR) is one of the most important perinatal disorders; it can increase perinatal mortality and morbidity, cause neurological sequelae in infants, and serve as the developmental origin of diseases such as hypertension and diabetes mellitus. Currently, the only strategy perinatologists can adopt is to determine the optimal delivery time for the fetus. If the pregnancy is terminated too late, in utero death can occur; if the pregnancy is terminated too early, prematurity could result in adverse sequelae [1,2]. No specific medications are known to prevent or treat FGR to date [3], although several candidates such as aspirin, heparin, statins, and sildenafil have been used [4,5,6,7,8,9,10,11].

Tadalafil is a specific and long-acting phosphodiesterase (PDE) 5 inhibitor that is normally used to treat pulmonary hypertension, erectile dysfunction, and prostate hypertrophy. In an earlier study, we found that tadalafil treatment increased fetal weight and relieved hypertension and proteinuria in a mouse model of L-NG-nitroarginine methyl ester (L-NAME)-induced FGR and preeclampsia [12]. Tadalafil dilated the maternal blood sinuses in the placenta, but did not have the same effects on fetal capillaries. It also increased placental growth factor production. Another study we performed using the same model revealed that tadalafil improved the hypoxic condition in the placenta and fetal brain, which had been induced by L-NAME administration [13].

We also retrospectively analyzed 11 FGR cases wherein tadalafil was administered, and compared their characteristics with those of 14 cases who received conventional management for FGR [14]. The women were matched for maternal age, parity, and gestational age (GA). This retrospective study showed that both fetal growth velocity from enrolment to birth and birth weight were significantly higher in the tadalafil group. The prevalence of respiratory distress syndrome was significantly lower in the tadalafil group. As the next step, a phase I trial was conducted to confirm the safety of tadalafil administration for FGR [15]. No severe adverse events were observed following the initiation of a daily tadalafil dose of 10 mg, 20 mg, or 40 mg.

Sildenafil, another PDE5 inhibitor has been studied worldwide for FGR treatment [8,9,10,11]. Animal studies and clinical studies with a small number of patients have shown promising results [16,17,18,19,20,21,22]. However, the recent Sildenafil Therapy In Dismal Prognosis Early-onset Fetal Growth Restriction (STRIDER) U.K. study reported no benefits of sildenafil for pregnancy prolongation, survival, or short-term neonatal outcomes [10]. Similar STRIDER studies performed in New Zealand and Australia also reported that sildenafil did not affect the proportion of pregnancies with an increase in fetal growth velocity [23]. The Dutch STRIDER trial discontinued participant recruitment after an interim analysis revealed increased incidence of persistent pulmonary hypertension of the newborn occurring with sildenafil compared to placebo [24]. The results of these studies led to the conclusion that sildenafil should no longer be recommended for early-onset FGR beyond its use in research studies where the participants have provided clear and informed consent.

We have already started a multicenter phase II trial in September 2016 on the efficacy and safety of tadalafil in fetuses with early-onset growth restriction: the TADAFER II trial, for which the protocol has already been published [25]. After the STRIDER U.K. paper was published in February 2018, our grant sponsor, the Japan Agency for Medical Research and Development, recommended that we cease recruiting new candidates for the TADAFER II trial by the end of March 2018 to reanalyze and verify the safety of tadalafil using the 89 patients who had participated in the study at that time.

Because the safety of using PDE5 inhibitors as therapeutic agents for FGR has been a problem worldwide, this paper primarily focuses on the safety assessments performed in the TADAFER II population.

## 2. Material and Methods

### 2.1. Trial Design and Study Population

This was a multicenter randomized controlled phase II trial. Twenty-one medical centers in Japan registered cases in this trial (all centers are described in Appendix A. The original plan was to enroll 140 fetuses with FGR from major medical centers in Japan. The study protocol was previously published [25].

FGR is defined as fetal growth less than 1.5 standard deviations below the mean estimated fetal body weight (EFBW) according to GA as calculated from ultrasonography, based on the Japanese standard [26]. Fetuses were randomized into two groups: one group received conventional treatment for FGR according to Japanese guidelines and the other group received once-daily administration of 20 mg tadalafil in addition to conventional treatment until delivery. A daily dose of 20 mg tadalafil was selected as it produced fewer maternal adverse events compared to the 40 mg dose, based on our experience in the phase I clinical trial [15]. We informed the participants orally and in writing of the possible side effects, including headache and facial flushing. A diary, in which the patients were instructed to record their daily adherence and adverse events, was provided to each participant at the time of registration.

The inclusion criteria were: (1) pregnant women aged ≥20 years, (2) EFBW less than 1.5 standard deviations of the mean EFBW for GA according to the Japanese standard curve, (3) GA between 20 + 0 and 33 + 6 weeks, (4) singleton pregnancy, and (6) signed written informed consent.

The exclusion criteria were: (1) antepartum fetal ultrasonography (including Doppler velocimetry) or fetal heart rate monitoring at enrollment indicating delivery; (2) history of allergy to tadalafil; (3) concurrent medications that could interact adversely with tadalafil; (4) relative contraindication to tadalafil treatment secondary to renal and/or liver disease; (5) relative contraindication to tadalafil secondary to uncontrolled arrhythmia, hypertension (blood pressure (BP) > 170/100 mmHg), or hypotension (BP < 80/40 mmHg); (6) fetus with suspected chromosomal anomaly and/or multiple congenital anomalies; (7) relative contraindication to tadalafil treatment secondary to retinitis pigmentosa, coagulation defect, active gastric and/or intestinal ulcer, or venous obstructive disease; and (9) determination by the investigator that inclusion was inappropriate.

This study was approved by the Institutional Review Board of Mie University Hospital on 25 August 2016 (approval no. 3041). The protocol was approved by the institutional review boards of all participating institutions as well. The study is registered at the University Hospital Medical Information Network (UMIN, registry number: 000023778).

### 2.2. Randomization

Patients that satisfied all inclusion criteria were deemed eligible for enrollment. Individual study sites were responsible for guiding the potential participants through the informed consent process, including those referred for treatment. Investigators entered eligible patients’ information into the Eligibility Confirmation Form on the trial website (Clinical Trial Data Management System). The data management system checked the form’s content before registering a patient. If a patient met all inclusion criteria without violating any of the exclusion criteria, the data management system registered and allocated the patient into one of two arms: conventional FGR management or once-daily tadalafil plus conventional management. Randomization was performed using a modified Pocock and Simon’s minimization method that considered the GA (<28 or ≥28 weeks) and facility of enrollment as an adjustment factor. The investigators were blinded to the allocation algorithm. All enrolled participants received treatment within 7 days of registration. 

### 2.3. Safety Evaluation

Neonatal and maternal adverse events were analyzed in addition to fetal, neonatal, and infant deaths six months after stopping the trial. Neonatal adverse events included major disease caused due to prematurity like persistent pulmonary hypertension of newborns, which was found to be a major problem in the Dutch STRIDER study. The details of the death cases were verified. For safety analysis, all the patients were included, except those who declined participation in the study.

### 2.4. Other Evaluation

Because the registered cases in this trial did not reach the planned number, the primary outcome and secondary outcome that we had set before the start of the trial in our protocol were analyzed using post hoc analysis and reference level. The contents analyzed as outcomes and included in this paper were examined according to our original protocol.

### 2.5. Criteria for Delivery

To minimize bias in the baseline fetal condition defined at enrollment, fetal indications for delivery were established. These indications were developed from the results of a multicenter survey [27] of very low birth weight infants in Japan in a network database that obtained data from 82 level III perinatal centers. The survey data included neonatal intensive care unit (NICU) infant survival rates grouped according to birthweight and GA at birth. Infant survival rates were divided into three groups: Zone 1, with an NICU infant survival rate <60%; Zone 2, with an NICU infant survival rate of 60%–95%; and Zone 3, with an NICU infant survival rate ≥95% (Figure 1). Each group had different delivery criteria based on Doppler umbilical arterial blood flow studies, non-stress tests, contraction stress tests, and biophysical profiles. The patients registered in the present study had their delivery times determined with reference to these criteria (Table 1).

### 2.6. Statistical Analysis

The safety analysis set population was defined as cases that registered in the conventional treatment group or were administered more than one dose of tadalafil in the tadalafil treatment group.

The outcomes that we initially set before the start of this trial, including the primary and secondary outcomes, were analyzed as a secondary analysis set. This set was defined to include cases that met the inclusion criteria, did not meet the exclusion criteria, and received protocol-defined treatments. Intrauterine fetal death cases were excluded from this population because actual birth weight data were not available. 

All statistical analyses were performed according to a pre-specified plan. The Data Coordinating Center in Mie University Hospital supported data management, statistical analysis, and study reporting. All analyses were performed using SAS version 9.4 (SAS Institute, Cary, NC, USA).

### 2.7. Funding

This work was supported by the Japan Agency for Medical Research and Development (AMED) as part of the Project for Baby and Infant in Research of Health and Development to Adolescent and Young Adults, by the Japan Society for the Promotion of Science KAKENHI (Grant Number 17K16846), and in part by the Takeda Science Foundation.

### 2.8. Participant and Public Involvement

Patients were not involved in the design of this study or in the recruitment or conduct of this study. The results of this study will be published on the homepage of the Mie University of Obstetrics and Gynecology. For randomized controlled trials, there is no requirement for the intervention being assessed by patients themselves. Patients and/or the public were not involved in the design or conduct of this trial.

## 3. Results

### 3.1. Study Profile

Eighty-nine women were recruited between September 2016 and March 2018 from 21 participating medical centers in Japan. We randomized 45 patients into the tadalafil group and 44 into the conventional treatment group (Figure 2). One patient declined participation after randomization in the tadalafil treatment group, and one patient declined in the conventional treatment group. We included 44 cases in the tadalafil treatment group and 43 cases in the conventional treatment group in the safety analysis. The median time from randomization to treatment was one (0–6 days) day, and the average was 0.8 ± 1.3 days. Thirty-six patients (82%) in the tadalafil treatment group received tadalafil the same day or one day after randomization. All patients received tadalafil until delivery except one, who stopped the treatment due to side effects and following the patient’s request within one week. The mean time from administration of tadalafil to termination of pregnancy was 46.3 ± 26.1 days.

There were no differences in the baseline characteristics in the safety analysis population between the tadalafil treatment and conventional groups (Table 2). 

### 3.2. Adverse Events 

Six months after stopping the trial, there were seven (16%) deaths in the conventional treatment group and one (2%) in the tadalafil group (fetal, neonatal, and infant deaths: 0, 1, and 0 vs. 4, 1, and 2, respectively; *p* = 0.03; Table 3). Four intrauterine fetal deaths occurred in the conventional group, ranging from 20 to 25 weeks GA at birth. One neonate in the conventional group was born at 25 weeks and died secondary to respiratory failure. One neonate in the tadalafil group was born at 23 weeks and died secondary to necrotizing enterocolitis. Two infant deaths in the conventional group resulted from sepsis (*n* = 1) and necrotizing enterocolitis (*n* = 1) (Table 4).

The rate of major neonatal adverse events was not different between the two groups (Table 5). Two cases of persistent pulmonary hypertension occurred in each group, in which no differences were found.

Maternal adverse events are shown in Table 6. Grade 1 headache, facial flushing, and nasal hemorrhage, which had no influence on patient quality of life, were frequently observed in the tadalafil treatment group, as previously reported [15]. One threatened uterine rupture (Grade 4) and two threatened premature deliveries (Grade 3) accounted for the severe adverse events in the conventional treatment group. Abruptio placenta occurred in two patients in the tadalafil treatment group (5%) and one in the conventional treatment group (2%; *p* > 0.99). No severe adverse event (>Grade 3) associated with tadalafil administration were observed in this study.

### 3.3. Other Evaluation

The outcomes we set in the TADAFER II protocol [16] were secondarily analyzed as a secondary analysis set (Figure 2). Four cases of intrauterine fetal death occurred in the conventional treatment group. Two patients in the tadalafil treatment group did not satisfy the inclusion criteria of EFBW 1.5 SD or less at the time of registration, although they were diagnosed with FGR prior to enrollment. One patient in the tadalafil treatment group turned out to have met the exclusion criteria (hypertension) after the registration was completed. One patient in the tadalafil treatment group underwent artificial abortion at 21 gestational weeks due to the patient’s choice. In total, 79 cases were included in the analysis of protocol-defined outcomes. None of the participants were lost to follow-up. 

The characteristics of participants in secondary analysis set are shown in Table 2. No differences were found between the groups.

Fetal growth velocity (FGV) in the secondary analysis set was not significantly different between the tadalafil and conventional treatment groups (14.5 ± 7.3 g/day vs. 12.9 ± 8.2 g/day, respectively; *p* = 0.37; Table 7). 

No differences were found in GA at birth, birthweight, cesarean section rate, neonatal APGAR scores (Appearance, Pulse, Grimace, Activity, Respiration score), and umbilical artery blood gas analyses in secondary analysis set (Table 8). In the analysis of all cases, we found no differences between the two groups for prolongation of GA (47.2 ± 27.2 days vs. 37.9 ± 24.1 days, respectively; *p* = 0.11). Figure 3 shows GA prolongation in Kaplan–Meier curves by GA at treatment initiation. Notably, in the post hoc analysis, limiting cases registered at <30 or <32 weeks GA and including fetal death cases (equal to safety analysis set), tadalafil significantly prolonged pregnancy compared with conventional treatment (<32 weeks: 52.4 ± 28.9 days vs. 36.8 ± 26.8 days, respectively, *p* = 0.03; <30 weeks: 55.0 ± 30.4 days vs. 37.0 ± 30.5 days, respectively, *p* = 0.04; Appendix A). 

The incidence of hypertensive disorders of pregnancy was not different between the groups (7% vs. 16%, respectively; *p* = 0.43); however, tadalafil seemed to delay the onset and decrease the occurrence of hypertensive disorders. FGVs two weeks after initiating protocol-defined treatments were significantly different between the conventional treatment and tadalafil groups (19.1 ± 9.5 g/day vs. 14.3 ± 8.6 g/day, respectively; *p* = 0.02). The fetal growth rate at two weeks after initiating protocol-defined treatments significantly increased in the tadalafil group compared with the conventional treatment group (2.5 ± 1.1 %/day vs. 1.8 ± 1.0%/day, respectively; *p* = 0.01). When limited to cases registered at <30 weeks GA, we found significant differences in GA at birth between the tadalafil and conventional treatment groups (37.07 weeks, interquartile range (IQR) 30.9–37.9 vs. 31.5 weeks, IQR 28.9–36.2, respectively; *p* = 0.048, Appendix A).

Although we analyzed maternal and fetal vessel blood flows (umbilical artery, middle cerebral artery, and uterine artery) according to the pulsatility index (PI), no significant differences were found in these scores between the groups (Table 9). The PI scores converted to MoM to exclude the influences of GA are also presented. There were no significant differences between the two groups. Additional analysis of the doppler study divided into the cases <32 gestational weeks and the cases ≥32 gestational weeks at registration are shown in Appendix A.

The treatment completion rate, defined by the percentage of enrolled patients who received the protocol-defined treatment for more than seven days, was not significantly different between the tadalafil and conventional treatment groups (100% vs. 92%, respectively; *p* = 0.12).

## 4. Discussion

The most important result of this safety evaluation study is the significant decrease in fetal, neonatal, and infantile deaths in the tadalafil group to the conventional treatment group: 1/44 (2%) vs. 7/43 (16%), respectively (*p* = 0.03). This contrasts with the results of the Dutch STRIDER trial with sildenafil treatment for early-onset FGR. The Dutch STRIDER trial stopped after interim analysis revealed that 11 newborns died due to persistent pulmonary hypertension of neonate (PPHN) in 93 patients in the sildenafil-treatment group, whereas no death from PPHN occurred in 90 cases in the control group [24].

What caused this difference? First, tadalafil is a different medicine from sildenafil. Walton RB et al. reported that sildenafil citrate improved preconstricted placental-fetal arterial perfusion in a human placental model, whereas tadalafil produced no response [28]. This indicates that tadalafil does not cross the human placental barrier, or is degraded by trophoblasts. We noted that tadalafil improved the width of the maternal blood sinuses in the placenta of preeclampsia model mice, but did not significantly change the fetal capillaries [12]. These facts convinced us that tadalafil functions only on the maternal side, not on the fetal side. Tadalafil is thought to work safely for fetuses and newborns, as it does not directly affect the fetus, contrary to the occurrence of neonatal persistent pulmonary hypertension in the Dutch STRIDER study. The second reason is that tadalafil is more stable than sildenafil, because tadalafil has a longer half-life (14 h vs. 4 h, respectively), is less susceptible to food intake, and has a more selective effect on PDE5.

There are several explanations why tadalafil decreased fetal and infant mortality. All deaths occurred in the cases born ≤28 weeks of GA, indicating that prematurity is an important factor in addition to FGR for death [29]. In our study, three cases with absent or reversed end-diastolic flow of the umbilical artery died in the conventional treatment group, whereas two cases with such abnormal flow survived in the tadalafil group. The post-hoc analysis revealed that tadalafil prolonged pregnancy for more than two weeks, with statistical significance, for the cases registered at <30 weeks of GA. GA at birth was also significantly prolonged to a median of 37.1 weeks from 31.5 weeks in the tadalafil group compared to the conventional treatment group. It seems unlikely that researchers intentionally tried to extend the pregnancy period with the expectation of efficacy of tadalafil, because fetal deaths would have increased in the tadalafil treatment group if investigators waited too long until the condition of the fetus worsened. We think that tadalafil-induced prolongation is the main cause for the decrease in fetal and infant mortality. 

In the STRIDER U.K. study, the subjects were in advanced stages of FGR with both size and Doppler abnormalities necessary for inclusion. GA at randomization was from 22 weeks and 0 days to 29 weeks and 6 days, which was much earlier compared to our study and focused on a much more severe type of FGR. In addition, they defined FGR as a fetus with abdominal circumference or estimated fetal weight below the 10th percentile using local charts and absent or reversed end-diastolic flow in the umbilical artery on Doppler velocimetry. As a result, the efficacy of sildenafil for these cases was not proven because of their severity. Although less severe cases were enrolled in the present study, the medians of approximately the second percentile of FGR cases for both groups were included, as shown in Table 2, which are very severe cases associate with perinatal mortality and morbidity. In the conventional group, seven cases of death (16%) occurred. This result confirmed that severe cases were included in this study.

Almost all the maternal adverse events in the tadalafil group were within Grade 1, transient, and did not need any medication, although headache (57%), facial flushing (36%), and nasal hemorrhage (16%) were more frequently observed in the tadalafil group compared to the conventional treatment group. Our unpublished data found a significantly low incidence of adverse effects in pregnant women compared to nonpregnant women. We speculate that this was contributed by an upregulated NO-cyclic guanosine monophosphate (GMP) system in pregnancy and accommodated a high cyclic GMP state induced by tadalafil. The incidence of neonatal adverse events, including pulmonary hypertension, was equal between the tadalafil and conventional treatment groups. These results warrant our progressing to further study to evaluate efficacy of tadalafil on FGR.

In the tadalafil treatment group, the number of cases of newly developed gestational hypertension and preeclampsia tended to decrease. Tadalafil may have a beneficial effect on reducing the incidence of gestational hypertension and preeclampsia. A nonsignificant trend of fewer cases of new-onset preeclampsia and preeclampsia as the delivery indication in women treated with sildenafil was also reported in the NZAus STRIDER trial [23]. Further investigation is warranted.

A major strength of our study is that we strictly set the criteria for the termination of pregnancy using Zones 1, 2, and 3 according to the neonatal mortality rate. This also minimized bias in terms of fetal baseline condition at enrollment in addition to the termination. We selected an open-label study design with a strict fetal management algorithm on the basis of the results from a multicenter Japanese survey instead of a placebo-controlled design due to operational challenges, including low acceptability by pregnant women in Japan. No treatment exists for FGR, and the best optimal termination by intense monitoring of the fetal condition is the standard management. This study focused on the safety of tadalafil for FGR as described above. Since this trial was prematurely stopped, the number of registered cases was less than planned number, and the study is an open-label design that has no placebo controls, the verification of tadalafil efficacy on early-onset FGR as therapeutic agent was not established. Now, we are planning a placebo-controlled, double blinded multicenter phase II study of tadalafil on early-onset FGR.

## 5. Conclusions

In conclusion, this trial showed that tadalafil decreased the fetal and infant deaths associated with FGR. This is thought to be contributed primarily by pregnancy prolongation. Administration of tadalafil was accompanied with transient low-grade maternal adverse effects like headache. Further studies are warranted to evaluate efficacy of the tadalafil on early-onset FGR.

## Figures and Tables

**Figure 1 jcm-08-00856-f001:**
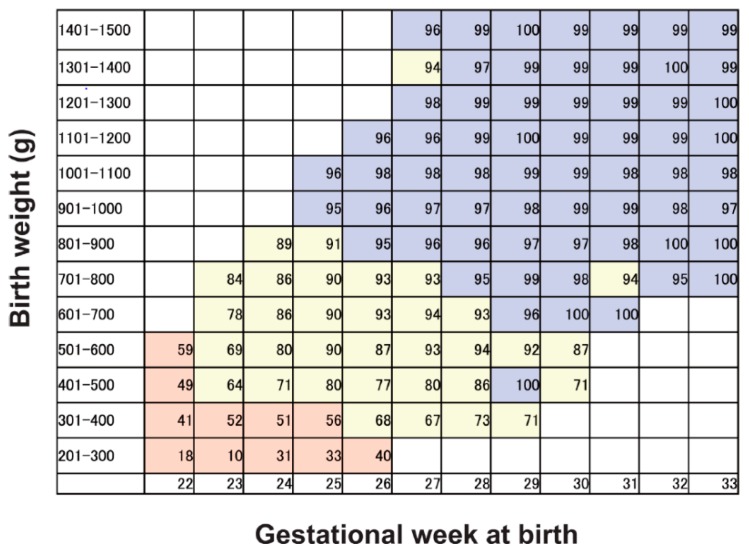
Infant survival rate in the neonatal intensive care unit (NICU) categorized according to birth weight and gestational week at birth (%). This figure was established on the basis of the results from the multicenter survey of very low birthweight infants in Japan using a network database. The survey data included infant survival rates in the NICU categorized by birth weight and gestational week at birth. The infant survival rate data acquired from the survey were preprocessed with the moving average method and divided into three groups. The first group was defined as Zone 1, where the infant survival rate in the NICU was less than 60% (pink background). The second group was defined as Zone 2, where the infant survival rate in the NICU ranged from 60% to 95% (yellow background). The third group was defined as Zone 3, where the infant survival rate in the NICU was 95% or higher (blue background).

**Figure 2 jcm-08-00856-f002:**
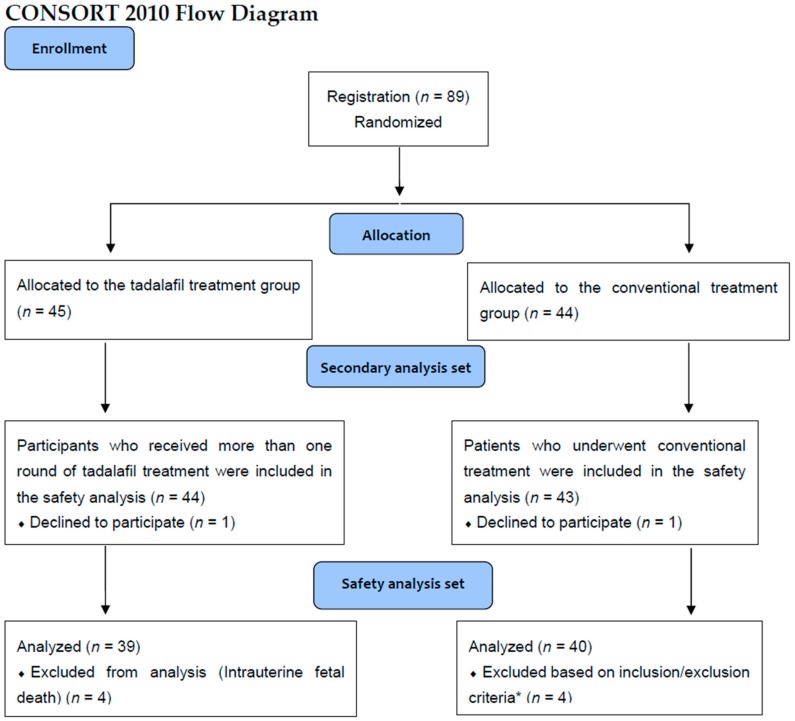
Flowchart patient of recruitment into this study group. * Two patients did not satisfy the inclusion criteria of EFW -1.5SD or less at the time of registration, although diagnosed with FGR prior to enrollment. One patient turned out to have met exclusion criteria (hypertension) at registration after registration was completed. One patient underwent medical termination of pregnancy.

**Figure 3 jcm-08-00856-f003:**
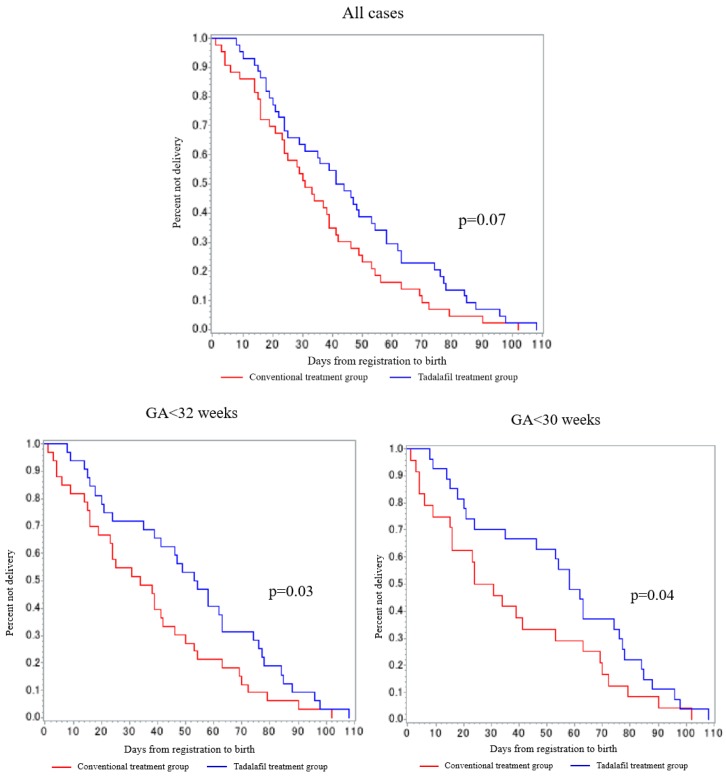
Prolongation of gestational weeks. Prolongation of GA was defined as days from the first day of protocol-defined treatment to birth. The figure above shows the prolongation of GA in Kaplan–Meier curves for each GA at treatment initiation (all cases, <32 weeks, <30 weeks), comparing binary data across two groups using the generalized Wilcoxon test.

**Table 1 jcm-08-00856-t001:** The fetal indications for delivery in the tadalafil for fetuses with early-onset growth restriction (TADAFER) phase II study.

Infant Survival Rate in the Neonatal Intensive Care Unit (NICU) (See Figure 1)	Fetal Indication for Delivery
Zone 1	Decide timing of delivery depending on available therapeutic measures at the NICU in each institute.
Zone 2	Consider delivery if at least one of three findings is made. 1. Reversed umbilical artery blood flow during diastole. 2. Score less than 4 on the fetal biophysical profile score. 3. Fetal heart rate patterns in the orange or red category for more than 30 min.
Zone 3	Consider delivery if at least one of the following five findings is made. 1. Reversed or absent umbilical artery blood flow during diastole, 2. Score less than 4 on the fetal biophysical profile score (score less than 6 on the fetal biophysical profile score if oligohydramnios is present). 3. Fetal heart rate patterns in the orange or red category for more than 30 min. 4. Positive contraction stress test. 5. Impaired fetal head circumference growth for more than 2 weeks,

**Table 2 jcm-08-00856-t002:** Patient characteristics at randomization.

	Tadalafil Treatment Group		Conventional Treatment Group	
	Safety Analysis Set	Secondary Analysis Set	Safety Analysis Set	Secondary Analysis Set
	(*n* = 44)	(*n* = 40)	(*n* = 43)	(*n* = 39)
Age (years)	34 (21–44)	33 (27–38)	34 (20–43)	34 (29–37)
Height (cm)	158 (147–170)	157 (153–160)	156 (144–168)	158 (155–162)
Weight (kg)	56 (45–91)	56 (52–64)	56 (44–75)	57 (51–61)
Smoking history	3 (7%)	3 (8%)	2 (5%)	2 (5%)
Nulliparous	21 (47%)	22 (55%)	26 (60%)	15 (38%)
Gestational age at registration (weeks)	29 (25–32)	28 (25–32)	28 (24.5–31)	29 (26–32)
Obstetrics complications				
Gestational hypertension	6 (13%)	6 (15%)	5 (12%)	3 (8%)
Preeclampsia	3 (7%)	1 (3%)	3 (7%)	3 (8%)
Gestational diabetes	5 (11%)	5 (13%)	0 (0%)	0 (0%)
Ultrasound sonographic finding				
Estimated fetal body weight (g)	887 (444–1349)	887 (520–1361)	809 (504–1348)	900 (620–1384)
Standard deviation of estimated fetal body weight	−2.3 (−2.8 to −1.9)	−2.3 (−2.8 to −2.0)	−2.2 (−2.9 to −2.0)	−2.1 (−2.6 to −1.9)
Percentile of estimated fetal body weight	2.1 (0.3–2.2)	2.1 (0.3–2.2)	2.2 (0.2–2.3)	2.2 (0.2–2.3)
Umbilical artery Doppler abnormal ^*1^	2 (5%)	2(5%)	3 (7%)	0 (0%)
Uterine artery Doppler abnormal ^*2^	12 (27%)	10(30%)	15 (35%)	11 (32%)
Maternal factors				
Systolic BP (mmHg)	120 (88–151)	114 (103–130)	114 (90–185)	120 (106–132)
Diastolic BP (mmHg)	73 (46–100)	71 (59–81)	71 (49–110)	73 (65–80)
Creatinine (mg/dL)	0.48 (0.31–0.9)	0.47(0·41–0·53)	0.47 (0.34–0.74)	0.47(0.40–0.53)
AST (IU/L)	17 (9–49)	16 (14–21)	16 (10–31)	16 (13–21)
ALT (IU/L)	11 (4–71)	12 (8–17)	12 (5–49)	11 (7–15)
Albumin (g/dL)	3.2 (2.2–4)	3.3 (3.0–3.5)	3.3 (2.1–4.0)	3.2 (3.0–3.5)
Platelet count (×10^4^/μL)	24 (9–35)	25.0 (22.2–28.1)	24 (9–37)	23.2 (19.4–26.2)

Data are reported as median (interquartile range), *n* (%). *^1^ End-diastolic flow absent or reversed; *^2^ Notching present; BP: Blood pressure; AST: Aspartate transaminase; ALT: Alanine transaminase.

**Table 3 jcm-08-00856-t003:** Fetal, neonatal, and infant deaths. Data are presented as *n* (%); NA: not available.

	Tadalafil Treatment Group (*n* = 44)	Conventional Treatment Group (*n* = 43)	*p* Value
Intrauterine fetal death	0 (0%)	4 (9%)	NA
Perinatal mortality	0 (0%)	5 (12%)	NA
Neonatal death	1 (2%)	1 (3%)	>0.99
Infant death (6 months after stopping trial)	0 (0%)	2 (4%)	NA
Total death	1 (2%)	7 (16%)	0.03

**Table 4 jcm-08-00856-t004:** Details of fetal, neonatal, and infant death cases.

Timing of Death	Allocation	Gestational Weeks at Registration	Gestational Weeks at Delivery	Treatment Period (Days)	Birth Weight (g)	Age in Days at the Time of Death (Days)	Cause of Death	Umbilical Artery Flow Absent or Reverse at Registration
Fetal death	Conventional treatment	23	25	10	328	NA	Placental abruption	+
Fetal death	Conventional treatment	20	20	4	<300	NA	Unknown	+
Fetal death	Conventional treatment	25	25	6	440	NA	Unknown	+
Fetal death	Conventional treatment	21	25	30	484	NA	Unknown	−
Neonatal death	Tadalafil treatment	21	23	16	317	17	Necrotizing enterocolitis, Sepsis	−
Neonatal death	Conventional treatment	24	25	7	440	0	Respiratory failure	−
Infant death	Conventional treatment	27	27	0	704	98	Sepsis, Renal failure	−
Infant death	Conventional treatment	28	32	25	730	167	Necrotizing enterocolitis	−

+: present; −: not present; NA: not available.

**Table 5 jcm-08-00856-t005:** Neonatal adverse events.

	Tadalafil Treatment Group (*n* = 44)	Conventional Treatment Group (*n* = 43)	Relative Risk	95% Confidence Interval	*p*-Value
Infant admitted to NICU	36 (82%)	33 (77%)	1.03	0.84–1.27	0.61
Oxygen dependency	16 (36%)	18 (42%)	0.82	0.49–1.37	0.66
Necrotizing enterocolitis	1 (2%)	2 (5%)	0.48	0.04–5.19	0.62
Retinopathy of prematurity	5 (11%)	4 (9%)	1.22	0.35–4.24	>0.99
Intraventricular hemorrhage	0 (0%)	1 (2%)	NA	NA	0.49
Periventricular leukomalacia	1 (2%)	0 (0%)	NA	NA	>0.99
Hypoxic ischemic encephalopathy	0 (0%)	0 (0%)	NA	NA	>0.99
Respiratory distress syndrome	9 (20%)	12 (28%)	0.67	0.32–1.41	0.46
Surfactant use	7 (16%)	11 (26%)	0.57	0.24–1.31	0.30
Chronic pulmonary disease	6 (14%)	3 (7%)	1.94	0.52–7.32	0.48
Persistent pulmonary hypertension of the newborn	2 (5%)	2 (5%)	0.97	0.14–6.62	>0.99
Patent ductus arteriosus	5 (11%)	5 (12%)	0.97	0.30–3.13	>0.99
Anemia of prematurity	12 (27%)	15 (35%)	0.73	0.39–1.36	0.49
Meconium plug syndrome	2(5%)	0(0%)	NA	NA	0.49

Data are *n* (%), NA: not available; NICU: neonatal intensive care unit.

**Table 6 jcm-08-00856-t006:** Maternal adverse events. NICU: Neonatal intensive care unit.

	Tadalafil Treatment Group (*n* = 44)				Conventional Treatment Group (*n* = 43)			
	Grade 1	Grade 2	Grade 3	Grade 4	Grade 1	Grade 2	Grade 3	Grade 4
Headache	25(57%)	1 (2%)	0 (0%)	0 (0%)	11 (26%)	0(0%)	0 (0%)	1 (2%)
Facial flushing	16 (36%)	0 (0%)	0 (0%)	0 (0%)	0 (0%)	0 (0%)	0 (0%)	0 (0%)
Palpitations	5 (11%)	0 (0%)	0 (0%)	0 (0%)	2 (5%)	0 (0%)	0 (0%)	0 (0%)
Anorexia	9 (20%)	1 (2%)	0 (0%)	0 (0%)	3 (7%)	0 (0%)	0 (0%)	0 (0%)
Nausea	8 (18%)	1 (2%)	0 (0%)	0 (0%)	3 (7%)	0 (0%)	0 (0%)	0 (0%)
Dizziness	6 (14%)	1 (2%)	0 (0%)	0 (0%)	2 (5%)	0 (0%)	0 (0%)	0 (0%)
Muscle pain	5 (11%)	1 (2%)	0 (0%)	0 (0%)	3 (7%)	0 (0%)	0 (0%)	0 (0%)
Nasal hemorrhage	7 (16%)	0 (0%)	0 (0%)	0 (0%)	0 (0%)	0 (0%)	0 (0%)	0 (0%)
Indigestion	1 (2%)	0 (0%)	0 (0%)	0 (0%)	1 (2%)	0 (0%)	0 (0%)	0 (0%)
Diarrhea	3 (7%)	0 (0%)	0 (0%)	0 (0%)	2 (5%)	0 (0%)	0 (0%)	0 (0%)
Breathing trouble	5 (11%)	0 (0%)	0 (0%)	0 (0%)	4 (9%)	0 (0%)	0 (0%)	0 (0%)

**Table 7 jcm-08-00856-t007:** Fetal growth velocity as secondary analysis. Data are reported as mean (standard deviation).

	Tadalafil Treatment Group (*n* = 40)	Conventional Treatment Group (*n* = 39)	*p*-Value
Fetal growth velocity (g/day)	14.5 (7.3)	12.9 (8.2)	0.37

**Table 8 jcm-08-00856-t008:** Outcome according to treatment as secondary analysis.

	Tadalafil Treatment Group (*n* = 40)	Conventional Treatment Group (*n* = 39)	*p* Value
Prolongation of GA (days)	47.2 (27.2)	37.9 (24.1)	0.11
GA at birth (weeks)	37.0 (33.9–37.9)	36.0 (31.0–37.6)	0.23
Birth weight (g)	1639 (615)	1548 (713)	0.55
Standard deviation of birth weight	−2.5 (−3.2 to −1.5)	−2.1 (−3.2 to −0.9)	0.76
Percentile of birth weight	0.6 (0.1–6.1)	1.8 (0.1–5.5)	0.81
Maximum vertical pocket (cm)	4.3 (3.2–5.3)	4.1 (3.2–5.2)	0.93
Onset of hypertensive disorders of pregnancy *, **			
1 week from the start of treatment	0 (0%)	1 (3%)	>0.99
2 weeks from the start of treatment	0 (0%)	3 (9%)	0.24
3 weeks from the start of treatment	0 (0%)	4 (13%)	0.11
4 weeks from the start of treatment	0 (0%)	4 (13%)	0.11
>4 weeks from the start of treatment	2 (7%)	5 (16%)	0.43
Obstetric complication			
Abruption of placenta	2 (5%)	1 (2%)	>0.99
Gestational diabetes	5 (11%)	0 (0%)	0.06
Cesarean section	29 (74%)	31 (79%)	0.79
Newborn’s sex			
Male	17 (43%)	18 (46%)	0.82
Female	23 (58%)	21 (54%)	0.82
Apgar score			
1 min	8 (7–8)	8 (5–8)	0.76
5 min	9 (9–9)	9 (8–9)	0.30
Umbilical cord blood gas analysis			
pH	7.284 (7.245–7.327)	7.288 (7.262–7.317)	0.66
BE	−4.3 (3.9)	−3.5 (3.6)	0.35
Intrauterine fetal death	0 (0%)	4 (9%)	0.06
Perinatal mortality	0 (0%)	4 (9%)	0.06
Neonatal death	1 (3%)	1 (3%)	>0.99
Use of aspirin	2 (5%)	1 (3%)	>0.99
Fetal growth velocity in the two weeks after the protocol-defined treatment (g/day)	19.1 (9.5)	14.3 (8.6)	0.02
Fetal growth rate in the two weeks after the protocol-defined treatment and from the first day of the protocol-defined treatment to birth (%/day)	2.4 (1.0)	1.8 (1.0)	0.01
Fetal growth rate from the first day of the protocol-defined treatment to birth (%/day)	1.9 (1.1)	1.5 (1.4)	0.14

Data are presented as median (interquartile range), *n* (%), or mean (standard deviation); * The number of cases is cumulative and shows the total number of cases at each time point. ** Hypertensive disorder of pregnancy was defined as newly developed gestational hypertension and preeclampsia after registration. Gestational hypertension or preeclampsia was defined according to the American College of Obstetricians and Gynecologists guidelines.

**Table 9 jcm-08-00856-t009:** Doppler study analysis of maternal or fetal vessels.

	Tadalafil Treatment Group (*n* = 40)	Conventional Treatment Group (*n* = 39)	*p*-Value
Umbilical artery			
PI at registration	1.03 (0.95–1.28)	1.14 (0.98–1.54)	0.43
MoM	1.03 (0.85–1.19)	1.09 (0.91–1.24)	0.74
PI 1 week from the start of treatment	0.98 (0.86–1.29)	1.09 (0.95–1.36)	0.18
MoM	0.94 (0.84–1.14)	1.09 (0.90–1.20)	0.50
PI 2 weeks from the start of treatment	0.66 (0.61–0.72)	0.66 (0.60–0.73)	0.72
MoM	1.05 (0.90–1.22)	1.01 (0.90–1.17)	0.71
PI 3 weeks from the start of treatment	0.63 (0.54–0.72)	0.64 (0.59–0.73)	0.37
MoM	1.01 (0.81–1.17)	1.01 (0.89–1.22)	0.93
Middle cerebral artery			
PI at registration	1.57 (1.28–1.96)	1.54 (1.15–1.89)	0.66
MoM	0.85 (0.70–1.05)	0.80 (0.69–0.92)	0.75
PI 1 week from the start of treatment	1.65 (1.37–2.06)	1.68 (1.44–2.14)	0.73
MoM	0.86 (0.77–1.06)	0.89 (0.78–1.07)	0.72
PI 2 weeks from the start of treatment	1.71 (1.29–1.85)	1.61 (1.31–1.82)	0.63
MoM	0.88 (0.71–0.98)	0.85 (0.70–0.95)	0.81
PI 3 weeks from the start of treatment	1.58 (1.48–1.98)	1.66 (1.46–2.02)	0.95
MoM	0.82 (0.74–1.01)	0.88 (0.70–1.07)	0.66
Uterine artery			
PI at registration	1.04 (0.88–1.49)	1.24 (0.80–1.82)	0.23
MoM	1.38 (1.12–2.03)	1.44 (1.16–1.92)	0.57
PI 1 week from the start of treatment	0.96 (0.76–1.42)	0.79 (0.70–1.24)	0.28
MoM	1.20 (1.01–1.48)	1.34 (1.04–1.97)	0.36
PI 2 weeks from the start of treatment	1.06 (0.70–1.47)	0.94 (0.73–1.35)	0.69
MoM	1.35 (1.00–1.89)	1.42 (1.08–1.86)	0.35
PI 3 weeks from the start of treatment	0.94 (0.75–1.22)	0.96 (0.62–1.55)	0.96
MoM	1.15 (0.91–2.04)	1.40 (1.14–1.78)	0.75

Data are presented as medians (interquartile range). MoM: multiple of median.

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
