# Peer review of "Safety Evaluation of Tadalafil Treatment for Fetuses with Early-Onset Growth Restriction (TADAFER): Results from the Phase II Trial"

_jcm, 2019, doi:10.3390/jcm8060856_

Reviewer 1 Report

This manuscript reports the results of important research into the possible treatment for tadalafil for treatment of severe growth restriction which uses data from a randomized controlled trial, but for which the study was terminated early due to safety concerns in studies of a different, but related medication.  The results suggest that tadalafil may improve outcomes, and do so safely. Further, the study had important design strengths. Overall, these results are important and should be reported.

Unfortunately, there are some concerns with the presentation and with the methods that may impact the study's conclusions.

Most importantly, there is some concern about immortal time bias, particularly in the safety analysis as the major findings seem to result from a prolongation of pregnancy in the treated group as the population was limited to those who received at least one dose of tadalafil (page 7) which was received within 7 days of randomization (page 4).  There is no indication of the timing of first dose in most cases.  However, since all analyses seem to show a survival benefit that is limited to the first week following randomization, this information is critical.  

There is some concern about the lack of masking of physicians and participants to treatment, particularly given the small sample size.  Therefore, there is some chance that there are differences between treatment groups. For example, page 19, the authors state that "It seems unlikely that researchers intentionally tried to extend the pregnancy period..." However, in the early treatment period researchers may have done that because they wanted to allow time for the drug to work - which would extend delivery time early  - which is where the effect was shown.

The manuscript is somewhat unclear about the specific methods, and which exclusions apply to which analytic data sets.  Of note is the fact that nearly 10% of the participants in the tadalafil group should not have been enrolled in the study.  This needs to be clarified as does the fact that 1 patient in each treatment arm "declined to participate".  Normally this would happen prior to randomization.

It is important to provide not just p-values, but also measures of effect (and measures of precision) such as relative risks and 95% confidence intervals to the observed results, and that the authors consider carefully the groups at risk of certain outcomes.  For example, ROP is unlikely to be assessed except in infants surviving past the neonatal period.  Although the numbers not meeting this threshold is small, it does, apparently differ by treatment group therefore, the results may be confounded. 

There is a typo in table 2 (preeclampsia).  Additionally, it would be important for the authors to clarify that in this table, these are baseline characteristics and discuss the proportion of women who developed some of these characteristics after randomization (for example gestational hypertension and/or preeclampsia). I also would prefer that table 7 be combined with table 2 for easy comparison of the subpopulation used in that analysis.

In line 262-264, the results presented directly contradicts the results presented in table 8.  I suspect that one, or the other, has been reversed.

Author Response

1. Most importantly, there is some concern about immortal time bias, particularly in the safety analysis as the major findings seem to result from a prolongation of pregnancy in the treated group as the population was limited to those who received at least one dose of tadalafil (page 7) which was received within 7 days of randomization (page 4).  There is no indication of the timing of first dose in most cases.  However, since all analyses seem to show a survival benefit that is limited to the first week following randomization, this information is critical.  

Our response: Thank you for your comment. We agree that the timing of the first dose of tadalafil is very important, as it could lead to bias, particularly in safety analysis. We analyzed the interval between the day of registration to the start of tadalafil administration. The median time was 1 day, indicating that tadalafil was administered early in most cases. We added the following text to the Results section:

“The median time from randomization to treatment was 1 (0–6) days, and the average was 0.8 ± 1.3 days. Thirty-six patients (82%) in the tadalafil treatment group received tadalafil the same day or 1 day after randomization. All patients received tadalafil until delivery except one, who stopped the treatment due to side effects and following the patient’s request within one week. The mean time from administration of tadalafil to termination of pregnancy was 46.3 ± 26.1 days.”

2. There is some concern about the lack of masking of physicians and participants to treatment, particularly given the small sample size.  Therefore, there is some chance that there are differences between treatment groups. For example, page 19, the authors state that "It seems unlikely that researchers intentionally tried to extend the pregnancy period..." However, in the early treatment period researchers may have done that because they wanted to allow time for the drug to work - which would extend delivery time early  - which is where the effect was shown.

Our response: We agree that the open-label design is a limitation of this study. In order to minimize the researcher’s intentional bias, we set the termination criteria. In addition, the involved organizers and physicians double-checked whether each case met the termination criteria. Although the reviewer said researchers might have intentionally tried to extend pregnancy, in this trial, we designed the protocol so as to minimize bias.

As described above, the median time from randomization to start of tadalafil (tadalafil treatment group) was only 1 day. Therefore, we were able to administer tadalafil at an early time point and minimize the intentional bias that might occur due to the physician’s expectation of tadalafil efficacy.

3. The manuscript is somewhat unclear about the specific methods, and which exclusions apply to which analytic data sets. Of note is the fact that nearly 10% of the participants in the tadalafil group should not have been enrolled in the study. This needs to be clarified as does the fact that 1 patient in each treatment arm "declined to participate". Normally this would happen prior to randomization.

Our response: Cases that received more than one round of tadalafil, and for whom the primary outcome in the protocol (fetal growth velocity) could be calculated, were included in the secondary analysis set. We described such population as the full analysis set in our original protocol. In total, 40 cases in the tadalafil treatment group and 39 cases in the conventional treatment group were included in the secondary analysis set in this paper.

Four cases in the tadalafil treatment group were excluded from the secondary analysis set. Among them, two patients did not satisfy the diagnosis of FGR at the time of registration, although FGR had been diagnosed prior to enrollment. Despite the previous diagnosis of FGR, the entering information on the trial website after registration did not meet the FGR criteria. One patient met one exclusion criterion (hypertension) at the registration, but this was noticed only after registration was completed. One physician did not notice one patient’s severe hypertension, and he entered the patient’s information on the trial website after registration. The clinical trial center pointed out this mistake already after tadalafil treatment had been started. In this patient, severe hypertension improved immediately, and pregnancy was not terminated at that time. One patient underwent artificial abortion at 21 gestational weeks due to the patient’s desire. Four cases of intrauterine fetal death in the conventional treatment group were excluded from the secondary analysis set because fetal growth velocity could not be calculated.

Because this trial had an open-label design, patients knew whether they were receiving tadalafil or no medication after randomization. Two patients declined to participate after randomization because they knew to which group they belonged. Even after registration, participants could decline to participate in the study; therefore, we were aware that drop-outs could occur after registration.

4. It is important to provide not just p-values, but also measures of effect (and measures of precision) such as relative risks and 95% confidence intervals to the observed results, and that the authors consider carefully the groups at risk of certain outcomes.  For example, ROP is unlikely to be assessed except in infants surviving past the neonatal period.  Although the numbers not meeting this threshold is small, it does, apparently differ by treatment group therefore, the results may be confounded. 

Our response: Thank you for your comment. We have added the relative risk and 95% confidence interval in Table 5.

5. There is a typo in table 2 (preeclampsia). Additionally, it would be important for the authors to clarify that in this table, these are baseline characteristics and discuss the proportion of women who developed some of these characteristics after randomization (for example gestational hypertension and/or preeclampsia). I also would prefer that table 7 be combined with table 2 for easy comparison of the subpopulation used in that analysis.

Our response: We have corrected the typo in Table 2 (preeclampsia). We added the following footnote in Table 8:

“Hypertensive disorder of pregnancy was defined as newly developed gestational hypertension and preeclampsia after registration. Gestational hypertension or preeclampsia was defined according to the American College of Obstetricians and Gynecologists guidelines.”

We also added the following text in the Discussion section:

“In the tadalafil treatment group, the number of cases of newly developed gestational hypertension and preeclampsia tended to be decreased. Tadalafil may have a beneficial effect on reducing the incidence of gestational hypertension and preeclampsia. A nonsignificant trend to fewer cases of new-onset preeclampsia and preeclampsia as the delivery indication in women treated with sildenafil was also reported in the NZAus STRIDER trial. [30] Further investigation is warranted.

The following reference are added,

 “30. Groom KM, McCowan LM, et al. STRIDER NZAus: a multicentre randomised controlled trial of sildenafil therapy in early-onset fetal growth restriction. BJOG. 2019 Feb 19. doi: 10.1111/1471-0528.15658.

Following the reviewer’s suggestion, we combined Table 2 and Table 7.

6. In line 262-264, the results presented directly contradicts the results presented in table 8.  I suspect that one, or the other, has been reversed.

Our response: Thank you for your comment. We revised the text related to Table 7 (previously Table 8) as follows:

“Fetal growth velocity (FGV) in the secondary analysis set was not significantly different between the tadalafil and conventional treatment groups (14.5 ± 7.3 g/day vs. 12.9 ± 8.2 g/day, respectively; p = 0.37; Table 7).”

Reviewer 2 Report

Thank you for asking me to review this interesting paper, a Phase II trial looking at maternal and fetal safety of Tadalafil for early onset fetal growth restriction. This is an important topic, given the absence of any effective treatment for established early onset fetal growth restriction, and the competing demands of avoiding prematurity with delivery yet avoiding asphyxial injury or stillbirth in utero. This is particularly important in light of the  negative findings of the STRIDER trials (UK and Aus/ NZ) and the safety signal form the Dutch trial with excess neonatal deaths due to pulmonary HT in the sildenafil group.

This is thus a timely publication, although does not really provide the hoped-for (and proposed) reassurance, given that the trial design and subjects included were so vastly to STRIDER. The authors propose the benefit seen in this trial relates to tadalafil being a better class of drug, but this may not be the only reason. Specifically, this trial recruited to 33+6,  whereas all STRIDER trials recruited until no later than 30 weeks and all ceased treatment by 32 weeks. The entry criteria for STRIDER were far more stringent, with both size and Doppler abnormalities necessary for inclusion, which increases the confidence that the population in question had true uteroplacental insufficiency. In contrast, this study took all patients with EFW<1.5SD (so just under the 10th centile) with no additional US or clinical criteria required. They are thus very different populations- this one being far lower in risk for adverse outcome. This is evidenced by the late gestational age at delivery, with the average GA in the treatment group 37 weeks, and control group 36 weeks. It is also evidenced by the very low perinatal mortality overall, compared to the 30% or so in the STRIDER UK study. I was unable to discern the actual numbers delivered at <32 and <30 weeks either form the text, tables or supp data (the groups reporting significant prolongation of pregnancy), but from the KM curves it looks like it may be well under half. The prolongation was not significantly different among those born at <26 week and <28 weeks, arguably the group where the greatest benefit is needed, and likely to be seen.

They authors report a reduction in death but no change in major morbidity with tadalafil. The finding is impressive but needs to be treated with caution. The stated primary and secondary outcomes (in the BMJ Open protocol) were negative.  The reduction in deaths was swayed by 2 late infant deaths, and 4 peri-viable in utero deaths that presumably were not offered delivery or resuscitation. Despite the standardised protocol for delivery (Fig 1/ Table 1), it would be helpful if criteria for not offering delivery were also clarified. While at relatively low risk of bias, the unblinded nature of the trial makes this important to clarify, as well as criteria for delivery in late gestation. Remarkably, despite being SGA, most of these pregnancies proceeded uneventfully to term, and there is potential for bias here. Although alluded to in the discussion, it is clear that most fetuses at advancing gestation were well, and not at high risk of stillbirth. Deferring delivery may have been attractive if there was clinician perception that women were taking a safe and effective drug to prolong gestation. 

Suggestions for Methods

1. To improve interpretation, suggest fetal and birthweight centiles should be reported 

2. Days of treatment should be reported, as well as average number of days from randomisation to treatment (could be up to 7)

3. All Doppler measurements need to be converted to MoM, given the huge impact of gestation on absolute values. They are meaningless without a gestational reference

Minor changes

The second last sentence (p. 17) needs restructuring

Table 2, typo for rate of preeclampsia (Can't be 22%)

Table 9 duplicate data (obstetric complication)

Line 260 should read, '...tadalafil functions...'

Author Response

1. I was unable to discern the actual numbers delivered at<32 and <30 weeks either form the text, tables or supp data (the groups reporting significant prolongation of pregnancy), but from the KM curves it looks like it may be well under half. 

Our response: Thank you for the comment. The number of cases with delivery at << span="">30 gestational weeks was 7/40 (17.5%) in the tadalafil treatment group and 11/39 (28.2%) in the conventional treatment group. The number of cases with delivery at << span="">32 gestational weeks was 8/40 (20.0%) in the tadalafil treatment group and 13/39 (33.3%) in the conventional treatment group.

2. They authors report a reduction in death but no change in major morbidity with tadalafil. The finding is impressive but needs to be treated with caution. The stated primary and secondary outcomes (in the BMJ Open protocol) were negative.  The reduction in deaths was swayed by 2 late infant deaths, and 4 peri-viable in utero deaths that presumably were not offered delivery or resuscitation. Despite the standardised protocol for delivery (Fig 1/ Table 1), it would be helpful if criteria for not offering delivery were also clarified. While at relatively low risk of bias, the unblinded nature of the trial makes this important to clarify, as well as criteria for delivery in late gestation. 

Our response: Most severe cases including intrauterine fetal death were categorized as Zone 1. This classification aids the decision on timing of delivery depending on the available therapeutic measures at the NICU in each institute, as shown in Figure 1 and Table 1. All institutions were tertiary centers with perinatal specialists; their judgement on delivery timing should be accurate and adequate.

Suggestions for Methods

1.     To improve interpretation, suggest fetal and birthweight centiles should be reported

Our response: Thank you for your suggestion. We have added the information of fetal and birthweight centiles in Table 2 and Table 8.

2.     Days of treatment should be reported, as well as average number of days from randomisation to treatment (could be up to 7)

Our response: The time from administration of tadalafil to termination of pregnancy was 46.3 ± 26.1 days. We added the following text in the Results section:

“The median time from randomization to treatment was 1 (0–6) days, and the average was 0.8 ± 1.3 days.”

3.   All Doppler measurements need to be converted to MoM, given the huge impact of gestation on absolute values. They are meaningless without a gestational reference.

Our response: Thank you for pointing out this important aspect. We used percentiles for Doppler analysis. We defined high PI in the umbilical artery as a value falling above the 95% percentile, and low PI in the middle cerebral artery as a value below the 5% percentile. We added the number of cases with high PI in the umbilical artery and low PI in the middle cerebral artery at the time of registration and after 1 week, 2 weeks, and 3 weeks from the start of treatment.

We added the following text to the Results section:

“High PI in the umbilical artery was defined as a value falling above the 95% percentile, and low PI in the middle cerebral artery was a value below the 5% percentile. The number of cases with high umbilical artery PI at registration/1 week/2 weeks/3 weeks from the start of treatment was 9/7/6/5 in the tadalafil treatment group and 16/12/9/8 in the conventional treatment group, respectively. The number of cases of low middle cerebral artery PI at registration/1 week/2 weeks/3 weeks from the start of treatment was 9/11/9/4 in the tadalafil treatment group and 16/10/11/7 in the conventional treatment group. There were no significant differences between the two groups.”

Minor change

1.  The second last sentence (p. 17) needs restructuring

Our response: We have revised the sentence as follows:

Additional analysis of the doppler study divided into the cases 32 gestational weeks and the cases 32 gestational weeks at registration were shown in Table S3.

2. Table 2, typo for rate of preeclampsia (Can't be 22%)

Our response: We revised the typo in Table 2.

3. Table 9 duplicate data (obstetric complication)

Our response: We deleted the duplicate data from Table 9.

4. Line 260 should read, '...tadalafil functions...'

Our response: We revised the sentence to “One patient in the tadalafil treatment group underwent artificial abortion at 21 gestational weeks due to the patient’s desire.”

Round  2

Reviewer 2 Report

My concerns are: Firstly, there has been no acknowledgement of the different populations (severity of FGR and gestational age), and why these findings are not particularly enlightening to assist with interpretation of the STRIDER data. The cohort is an unusual group. Despite being very SGA, the outcomes suggest they are largely low risk fetuses- born at term with normal pH- although the conventional group appeared to have more genuine uteroplacental insufficiency (if the proportion with abN dopplers is used as a surrogate marker).

Secondly, the decision for delivery for the 3/4 intrauterine fetal deaths at 25 weeks in the conventional treatment group is still poorly explained- and this mediated much of the proposed benefit. Given they had abN Dopplers, it appears they should have been considered for delivery, by the suggested criteria. If this was a blinded trial, this would be less concerning- but it raises a concern regarding potential bias.

The Doppler data is still poorly presented. Some clarification has been given regarding the proportion with abN dopplers, and indeed these suggest the conventional treatment group were worse at recruitment. Presented as they are, 

high umbilical artery PI at registration/1 week/2 weeks/3 weeks from the start of treatment was 9/7/6/5 in the tadalafil treatment group and 16/12/9/8 in the conventional treatment group, respectivelyThe number of cases of low middle cerebral artery PI at registration/1 week/2 weeks/3 weeks from the start of treatment was 9/11/9/4 in the tadalafil treatment group and 16/10/11/7 in the conventional treatment group 

the differences will clearly not be significant, but they appear overall to have more severe uteroplacental insufficiency. Continuing to provide absolute numbers in Table 9 (rather than MoMs) means it is difficult to reliably interpret the differences between the groups. It is surprising that the numbers with abN dopplers wasn't better balanced by randomisation. 

There is no neonatal data on vascular outcomes (echos- including pulmonary pressures etc) which would be hugely relevant if this paper is proposing to provide a safety (rather than an efficacy) signal- and this would be very reassuring. If the authors have such data, I would suggest they include it in further submissions.

Author Response

1. Firstly, there has been no acknowledgement of the different populations (severity of FGR and gestational age), and why these findings are not particularly enlightening to assist with interpretation of the STRIDER data. The cohort is an unusual group. Despite being very SGA, the outcomes suggest they are largely low risk fetuses- born at term with normal pH- although the conventional group appeared to have more genuine uteroplacental insufficiency (if the proportion with abN dopplers is used as a surrogate marker).

Our response: We sincerely appreciate this reviewer’s comment.

We have added the following sentence in the discussion section to acknowledge the differences in the population of the STRIDER trial and that of our study,

“In the STRIDER UK study, the subjects were in advanced stages of FGR with both size and Doppler abnormalities necessary for inclusion. GA at randomization was from 22 weeks and 0 day to 29 weeks and 6 days, which was much earlier compared to our study and focused on a much more severe type of FGR. In addition, they defined FGR as a fetus with abdominal circumference or estimated fetal weight below the tenth percentile using local charts and absent or reversed end-diastolic flow in the umbilical artery on Doppler velocimetry. As a result, the efficacy of sildenafil for these cases was not proved because of their severity. Although less severe cases were enrolled in the present study, the median approximately 2 percentile FGR cases both groups were included, as shown in Table 2, which are very severe cases associate with perinatal mortality, and morbidity. Actually, in the conventional group, 7 cases of death (16%) occurred. This result confirmed that severe cases were included in this study.

2. Secondly, the decision for delivery for the 3/4 intrauterine fetal deaths at 25 weeks in the conventional treatment group is still poorly explained- and this mediated much of the proposed benefit. Given they had abN Dopplers, it appears they should have been considered for delivery, by the suggested criteria. If this was a blinded trial, this would be less concerning- but it raises a concern regarding potential bias.

Our response: Thank you for the reviewer’s comment.

The birth weights of death cases were added in table 4 to help reader’s better understandings. The three cases the reviewer mentioned were actually judged to be zone 1 in the delivery criteria at each facility. Therefore, researchers in each facility decided not to conduct delivery, because its delivery indication was depending on available therapeutic measures at the NICU in each institute (Figure 1, Table 1).

3.  Continuing to provide absolute numbers in Table 9 (rather than MoMs) means it is difficult to reliably interpret the differences between the groups. 

Our response: Thank you for the reviewer’s comment. We converted all Doppler measurements to MoM and added to Table 1. No differences were found between two groups.

We added the following sentences in the result section,

The PI scores converted to MoM in order to exclude the influences of GA were also presented. There were no significant differences between the two groups.”

4. There is no neonatal data on vascular outcomes (echos- including pulmonary pressures etc) which would be hugely relevant if this paper is proposing to provide a safety (rather than an efficacy) signal- and this would be very reassuring. If the authors have such data, I would suggest they include it in further submissions. 

Our response: We agree to the reviewer’s comment, but we have no data for following the reviewer’s suggestion.